# LLM2Token: Distilling Large Language Models into Task-Specific Tokenizer

## Abstract

We present LLM to Tokenizer, a new distrilling method that preserves the prior knowledge from LLMs in the form of tokenizer, diverging from neural network based methods. This simple, intuitive method allows strong performance even only using one-hot encoding and a simgle-layer logistic regression. Based on the generated tokenizer, without any pre-training, surpressing GPT-2 with just 0.01% parameters. On event recognition tasks, the L2T method achieves an F1 score of 0.408 while using only 0.1% of the parameters compared to previous models. We also observed that stronger foundation models lead to improved tokenizer performance. And long tokenizers can harm the performance since the capacity of single-layer logistic regression is limited. This demonstrates a zero-shot capability of LLMs–through training on internet-scale corpora, they can recognize words that are important for specific tasks. We released all models, codes and the dataset to promote the furture exploration.

## 1 Introduction

Large language models (LLMs) have demonstrated unprecedented capabilities, leveraging vast knowledge acquired during pre-training to achieve state-of-the-art performance on a wide range of tasks (Brown et al., 2020; Touvron et al., 2023). Despite their success, their practical deployment is often hindered by immense computational and memory requirements, making inference costly and fine-tuning prohibitively resource-intensive (Chowdhery et al., 2023). In sight of this, knowledge distillation has emerged as a prominent technique to compress these massive models into smaller, more efficient counterparts (Guo et al., 2021). Broadly, distillation approaches fall into two categories: (1) response-based methods, which train a student to mimic the final output probabilities of the teacher LLM, and (2) feature-based methods that align the student's intermediate hidden states with those of the teacher (Sanh et al., 2019; Jiao et al., 2019).

In this paper, we aim to explore an entirely new paradigm that diverges from these conventional neural network-based distillation methods. Concretely, current distillation techniques typically follow the teacher-student paradigm (Hinton et al., 2015). The basic idea is to first select a smaller neural network architecture for the student model, and then train it to minimize a loss function that measures the divergence between its predictions and the teacher LLM's outputs. The common objective for this process is to make the student model, S, effectively replicate the behavior of the teacher, T, on a given dataset. Additionally, a standard cross-entropy loss against the ground-truth labels is often incorporated, acting as a constraint to ensure task-specific accuracy.

Despite their success, the current paradigm faces a critical limitation: it distills the LLM's knowledge into another, albeit smaller, neural network. This process inherently suffers from a loss of fidelity, as the student model's limited capacity struggles to fully capture the nuanced knowledge embedded within the teacher LLM's billions of parameters. This capacity mismatch often leads to a significant performance drop. Worse still, the student model still requires a costly and complex training phase, and its architecture remains a "black box," making the transferred knowledge difficult to interpret. The fundamental reliance on a neural network student curtails the potential for creating truly lightweight and efficient models.

To address these flaws, we instead propose distilling the LLM's knowledge not into a network's weights, but into a symbolic and interpretable representation: a task-specific tokenizer. We introduce a new paradigm where the LLM's role is not to teach a smaller model what to predict, but rather

to identify what features are important for a given task. Our method leverages the LLM's vast prior knowledge to generate a highly compact and optimized vocabulary of tokens that are most salient for the downstream task. By using this generated tokenizer, we can represent input text in a sparse and highly informative way. This allows an extremely simple model, such as a single-layer logistic regression with one-hot encoding, to achieve surprisingly strong performance without any pre-training. This approach effectively transfers the LLM's understanding of task-relevant semantics into the tokenization step itself, radically simplifying the downstream classifier. In a nutshell, we term the method as LLM2Token (L2T), a simple yet effective approach that distills LLMs into task-specific tokenizers.

To validate the effectiveness of our method, we conducted extensive experiments on event recognition tasks. The results show that, compared to previous models, L2T can achieve a competitive F1 score of 0.408 while using only 1% of the parameters and surprisingly surpassing the performance of GPT-3.5. This is achieved using just a single-layer logistic regression model that requires no pre-training. Furthermore, we empirically verified that stronger foundation models yield better-performing tokenizers, demonstrating a novel zero-shot capability of LLMs: their ability to discern task-critical lexical features from their vast pre-training. This highlights L2T's crucial role in creating efficient and interpretable models, enabling broader applications and future advancements in the field. All models and codes have been released to promote further exploration.

## 2 Preliminary

### 2.1 Autoregressive Language Model

An autoregressive large language model (LLM) generates sequences by predicting subsequent tokens $x$ conditioned on previously generated tokens. The internal representation of token $x$ at a given layer $l$, represented by the hidden state $h^l$, is computed through the following formulation:

$$h^l = h^{l-1} + a^l + m^l, \quad m^l = W_{\text{out}}^l, \sigma(W_{\text{in}}^l, \gamma(h^{l-1} + a^l), ),  \tag{1}$$

Here, $a^l$ corresponds to the attention block's output while $m^l$ denotes the contribution from the feed-forward network (FFN) layer at depth $l$. The FFN transformation involves two weight matrices: $W_{\text{in}}^l$ for the input projection and $W_{\text{out}}^l$ for the output projection. The function $\sigma$ represents the non-linear activation operation, and $\gamma$ indicates layer normalization applied to stabilize the hidden representations. In this formulation, consistent with the approach in Meng et al. (2022), we present the attention and FFN components as parallel computations rather than sequential operations, which provides a clearer view of their individual contributions to the final hidden state.

It is worth noting that $W_{\text{out}}^l$ within FFN layers is often interpreted as a linear associative memory, functioning as key-value storage for information retrieval (Geva et al., 2021). Specifically, if the knowledge stored in LLMs is formalized as $(s, r, o)$ — representing subject $s$, relation $r$, and object $o$ (e.g., $s$ = "The latest Olympic Game", $r$ = "was held in", $o$ = "Paris") — $W_{\text{out}}^l$ associates a set of input keys $k$ encoding $(s, r)$ with corresponding values $v$ encoding $(o)$. That is,

$$\underbrace{m^l}_{v} = W_{\text{out}}^l \underbrace{\sigma(W_{\text{in}}^l \gamma(h^{l-1} + a^l))}_{k}. \tag{2}$$

This interpretation has inspired most model editing methods to modify the FFN layers for knowledge updates (Hase et al., 2023; Li et al., 2024; Hu et al., 2024). For simplicity, we use $W$ to refer to $W_{\text{out}}^l$ in the following sections.

### 2.2 Knowledge Distillation

Knowledge distillation is a model compression technique aimed at transferring knowledge from a large, complex model (the "teacher," $\mathcal{T}$) to a smaller, more efficient model (the "student," $\mathcal{S}$) (Hinton et al., 2015). The core idea is to train the student to mimic the teacher's behavior. Instead of only learning from hard labels (e.g., one-hot vectors), the student is also trained on the "soft" probability distributions produced by the teacher. These soft targets contain richer information about the relationships between classes.

The standard distillation objective combines a distillation loss with a task-specific loss:

$$\mathcal{L} = \alpha \mathcal{L}_{\text{Distill}}(\sigma(\mathbf{z}_\mathcal{T}/\tau), \sigma(\mathbf{z}_\mathcal{S}/\tau)) + (1 - \alpha)\mathcal{L}_{\text{Task}}(\mathbf{y}, \sigma(\mathbf{z}_\mathcal{S})) \qquad (3)$$

where $\mathbf{z}_\mathcal{T}$ and $\mathbf{z}_\mathcal{S}$ are the logits from the teacher and student models, respectively. $\sigma$ is the softmax function, $\tau$ is a temperature parameter that softens the distributions, $\mathbf{y}$ are the ground-truth labels, and $\alpha$ is a weighting hyperparameter. Conventional methods distill knowledge into another neural network, which we argue is a fundamental limitation.

## 2.3 FEATURE SELECTION FOR TEXT CLASSIFICATION

Feature selection is the process of selecting a subset of relevant features (e.g., words, n-grams) from a high-dimensional vocabulary to build a learning model. Its primary goals are to improve model performance by removing irrelevant or noisy features, reduce computational complexity, and enhance model interpretability. Traditional methods for feature selection in NLP can be broadly categorized as:

- **Filter Methods:** These methods rank features based on statistical metrics computed from the data, independent of the classifier. Common metrics include the Chi-squared ($\chi^2$) test, Information Gain, and Term Frequency-Inverse Document Frequency (TF-IDF).

- **Embedded Methods:** In this approach, feature selection is integrated into the model training process. A classic example is L1 regularization (Lasso), which adds a penalty proportional to the absolute value of the weights, forcing the weights of less important features to become exactly zero.

These methods are purely data-driven, relying on the statistical properties of the training corpus. In contrast, L2T leverages the vast, pre-existing world knowledge encoded within an LLM to perform feature selection in a zero-shot or few-shot manner.

## 3 METHOD

We introduce LLM2Token (L2T), a novel knowledge distillation paradigm that distills a large language model's (LLM) knowledge into a task-specific tokenizer rather than a smaller neural network. This approach fundamentally shifts the complexity from the model architecture to the feature representation itself. The process involves two primary stages: (1) leveraging an LLM to generate a compact and highly informative vocabulary for a specific task, and (2) training a simple, interpretable downstream classifier on the resulting sparse representations. Figure 1 provides a high-level overview of the L2T workflow.

### 3.1 STAGE 1: LLM-POWERED TOKENIZER GENERATION

The core innovation of L2T lies in harnessing the extensive prior knowledge of an LLM to perform zero-shot feature selection. Instead of relying on statistical methods that require a large labeled dataset to compute feature relevance (e.g., chi-squared or TF-IDF variants), we directly query the LLM to identify the tokens most salient for the task.

**Problem Formulation**   Given a classification task $T$ with a label set $\mathcal{Y}$ (e.g., binary classification, $\mathcal{Y} = \{0, 1\}$), our goal is to construct a task-specific vocabulary (tokenizer) $V_T = \{w_1, w_2, \ldots, w_k\}$ of size $k$. This vocabulary should contain tokens that are maximally discriminative for distinguishing between the classes in $\mathcal{Y}$.

**Prompting Strategy**   We design a structured prompt to guide the LLM in generating the desired vocabulary. The prompt is composed of three key components:

- **Role and Task Definition:** We instruct the LLM to act as an expert in linguistics and data science. We clearly define the task, including a description of the positive and negative classes.

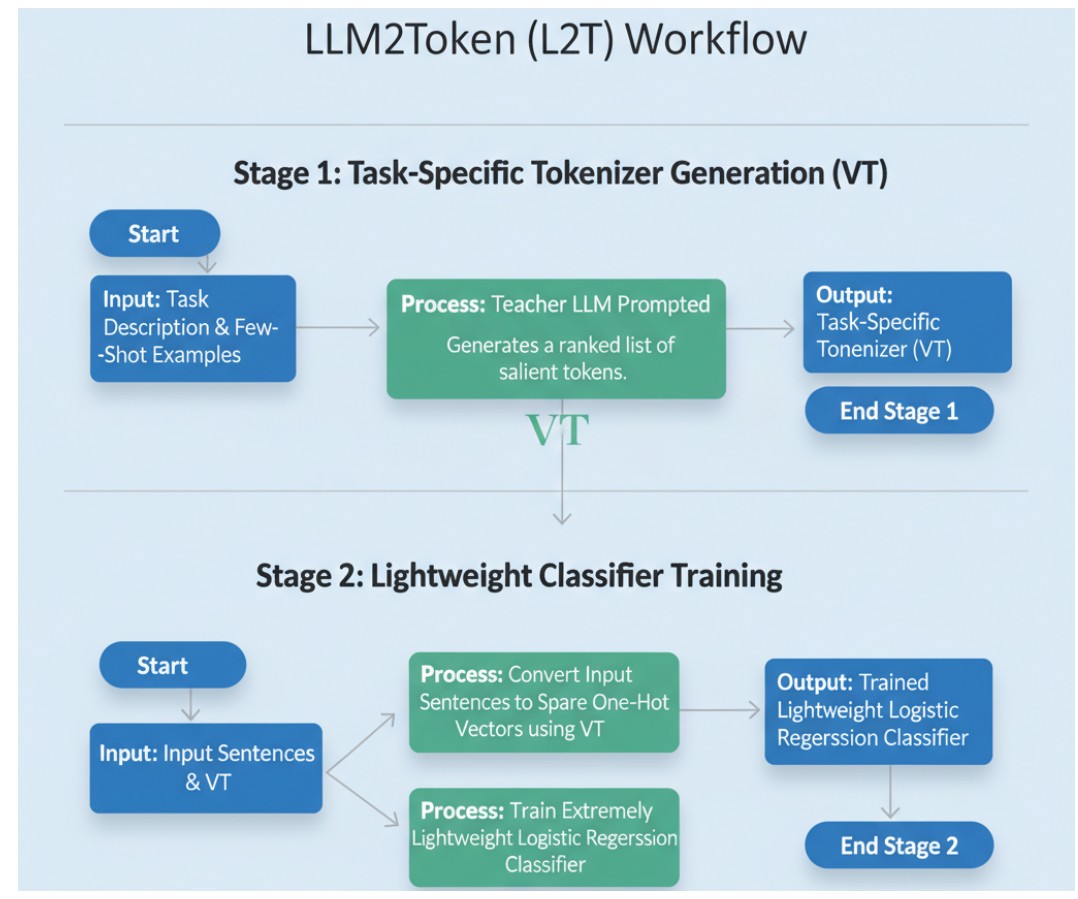

Figure 1: The LLM2Token (L2T) workflow. In Stage 1, a teacher LLM is prompted with the task description and few-shot examples to generate a ranked list of salient tokens, forming a task-specific tokenizer ($V_T$). In Stage 2, input sentences are converted into sparse one-hot vectors using $V_T$, which are then used to train an extremely lightweight logistic regression classifier.

- **Few-Shot Examples (Optional but Recommended):** We provide a small number of representative positive and negative examples. This helps to ground the LLM's understanding of the task's nuances and data distribution.

- **Output Instruction:** We explicitly ask the LLM to generate a ranked list of words and phrases that are highly indicative of the positive class. The instruction emphasizes the principle of "abnormal frequency"—identifying tokens that are disproportionately common in positive examples compared to general language or negative examples.

A template for our prompting strategy is shown in Figure 2. The LLM's response, a ranked list of tokens, is then truncated to the top-$k$ most relevant tokens to form our final task-specific tokenizer, $V_T$. The vocabulary size $k$ is a hyperparameter that balances feature richness and model simplicity.

### 3.2 STAGE 2: TEXT REPRESENTATION AND CLASSIFICATION

With the task-specific tokenizer $V_T$ generated, the subsequent stages are designed for maximum simplicity, efficiency, and interpretability.

**One-Hot Feature Representation** For any given input sentence $S$, we represent it as a $k$-dimensional binary vector $\mathbf{x} \in \{0, 1\}^k$. The representation is generated based on the presence

**System Prompt:** You are an expert in linguistics and data science. Your goal is to identify the most discriminative words for a text classification task.
**User Prompt:** I need to build a classifier to determine if a sentence is about "[TASK DESCRIPTION FOR POSITIVE CLASS]". I will provide you with a few examples. Based on these, please generate a ranked list of the top [k] words or short phrases that are the strongest indicators of the positive class. Focus on tokens that appear frequently in positive sentences but are rare in negative ones or in general text.
**Examples:** - [Example sentence t with positive/negative label]
**User Prompt:** Please provide a ranked list of the top [k] most indicative tokens for the "[POSITIVE/NEGATIVE]" class.

Figure 2: The prompt template used to guide the LLM in generating the task-specific tokenizer.

or absence of tokens from $V_T$ in the sentence. The $j$-th element of the vector, $x_j$, is defined as:

$$x_j = \begin{cases} 1 & \text{if } w_j \in S \\ 0 & \text{otherwise} \end{cases} \quad \text{for } j = 1, \ldots, k \tag{4}$$

where $w_j$ is the $j$-th token in our task-specific tokenizer $V_T$. This one-hot encoding approach is extremely simple and carries no additional information, thereby avoiding any influence from prior knowledge in pre-trained text embeddings on the results. If the task can be successfully completed using only one-hot encoding, it demonstrates that the LLM-generated tokenizer indeed contains task-relevant information.

**Downstream Classifier: Logistic Regression**  We employ a single-layer logistic regression model as our downstream classifier. This choice is deliberate, as its simplicity ensures that the performance gains are almost entirely attributable to the quality of the L2T-generated features rather than complex model interactions. The model predicts the probability of the positive class as:

$$P(y = 1|\mathbf{x}) = \hat{y} = \sigma(\mathbf{w}^T \mathbf{x} + b) = \frac{1}{1 + e^{-(\mathbf{w}^T \mathbf{x} + b)}} \tag{5}$$

where $\mathbf{w} \in \mathbb{R}^k$ is the weight vector, $b \in \mathbb{R}$ is the bias term, and $\sigma(\cdot)$ is the sigmoid function. The model is trained by minimizing the binary cross-entropy loss over the training dataset $D_{train}$:

$$\mathcal{L}(\mathbf{w}, b) = -\frac{1}{|D_{train}|} \sum_{(\mathbf{x}_i, y_i) \in D_{train}} [y_i \log(\hat{y}_i) + (1 - y_i) \log(1 - \hat{y}_i)] \tag{6}$$

The training is performed using standard optimization algorithms Adam (Kingma & Ba, 2014), without any need for pre-training.

**Advantages of the L2T Framework**  This two-stage methodology offers several key advantages over traditional distillation approaches:

- **Efficiency:** The final classifier is a simple logistic regression model with only $k + 1$ parameters, enabling near-instantaneous training and inference.
- **Interpretability:** The learned weights in the vector $\mathbf{w}$ directly correspond to the importance of each token in $V_T$ for the classification decision, making the model a "white box".
- **Decoupled Knowledge Transfer:** The LLM's knowledge is transferred once during the creation of the tokenizer. This static artifact, $V_T$, can then be used to train simple models on various datasets or in resource-constrained environments without needing further access to the LLM.

## 4 EXPERIMENT

### 4.1 IMPLEMENTATION DETAILS

We evaluate LLM2Token (L2T) on event recognition tasks using the People's Daily corpus (January 1998), containing 23,268 annotated sentences with part-of-speech tags following the 2003 corpus

specification. We focus on two binary classification tasks: (1) Meeting/Reception detection and (2) Inspection/Examination detection. The dataset is split into 60% training, 20% validation, and 20% test sets, with positive examples comprising less than 10% of the data. To prevent overfitting, we partition the data based on time. A model that performs well on the later test set demonstrates its ability to predict future events, not just memorize the training data.

For implementation, we use both scikit-learn (L-BFGS solver) and PyTorch with Adam optimizer. Learning rates range from $1 \times 10^{-6}$ to 0.1, training for up to 150,000 epochs on an RTX 3090 GPU. We apply StandardScaler normalization and employ one-hot encoding for text representation. The primary evaluation metric is F1-score: $F1 = \frac{2 \times \text{recall} \times \text{precision}}{\text{recall} + \text{precision}}$.

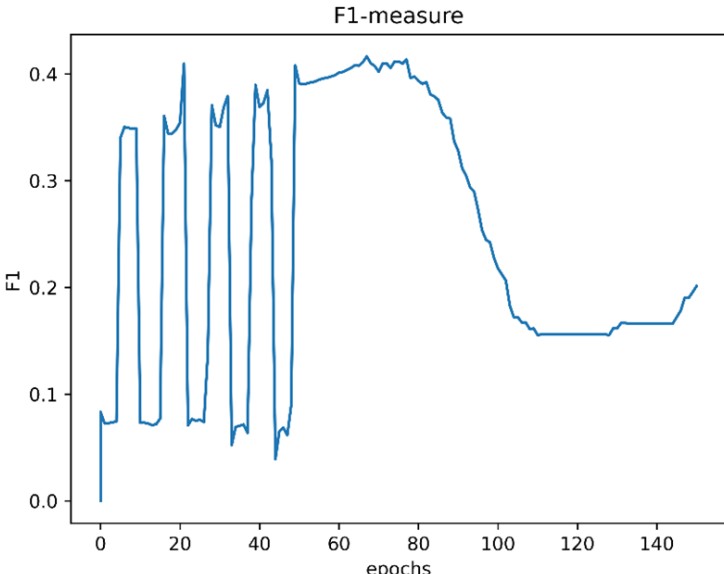

Figure 3: The training process of task 1. We change the learning rate to 1e-6 on 47 epoch. What surprising us is we get 0.4416 on val set and maxium of 0.5397 on test set. This shows that our dynamic changing method is work well.

## 4.2 BENCHMARKING RESULTS

Table 1: Performance comparison of different feature selection methods on event recognition tasks. The best and second best results are highlighted in red and blue.

| 2*Method | 2*Vocab Size | Task 1 (Meeting) | | Task 2 (Inspection) | |
|---|---|---|---|---|---|
| | | Val F1↑ | Test F1↑ | Val F1↑ | Test F1↑ |
| Manual Selection | 4,150 | 0.283 | 0.321 | 0.194 | 0.270 |
| Chi-square | 5,000 | 0.252 | - | - | - |
| Linear Model Fitting | 20,344 | 0.223 | 0.259 | - | - |
| **L2T (Ours)** | 4,709 | 0.382 | 0.459 | - | - |
| **L2T (Ours)** | 1,847 | **0.442** | **0.540** | - | - |
| **L2T (Ours)** | 1,598 | 0.405 | 0.512 | - | - |
| **L2T (Ours)** | 1,341 | - | - | **0.342** | **0.491** |
| **L2T (Ours)** | 620 | - | - | 0.314 | 0.489 |

The experimental results demonstrate that L2T with abnormal frequency selection significantly outperforms traditional methods. On Task 1, L2T achieves a test F1 of 0.540 with 1,847 words, representing a 68.2% improvement over manual selection. The optimal vocabulary size varies by task: approximately 1,800 words for Task 1 and 1,300 words for Task 2. Notably, linear model fitting with

20,344 features severely overfits (0.988 F1 on training, 0.259 on test), highlighting the importance of appropriate feature selection.

### 4.3 ABLATION STUDY

Figure 4: The pure training process without any specific tricks. This figure shows that our LLM generated features works quite well.

**Impact of Vocabulary Size.** We investigate how vocabulary size affects model performance, as shown in Table 2.

Table 2: Effect of vocabulary size on Task 1 performance. The best results are highlighted in red.

| Vocabulary Size | Val F1↑ | Test F1↑ | Training Behavior |
|---|---|---|---|
| 620 | - | - | Underfitting |
| 1,598 | 0.405 | 0.512 | Good generalization |
| 1,847 | **0.442** | **0.540** | Optimal |
| 4,709 | 0.382 | 0.459 | Slight overfitting |

The results reveal a sweet spot around 1,500-2,000 words. Smaller vocabularies lack discriminative power while larger ones introduce noise and increase overfitting risk due to the limited capacity of single-layer logistic regression.

**Optimization Strategies.** Table 3 presents the impact of different optimization techniques on model performance.

Table 3: Comparison of optimization strategies on validation set performance.

| Configuration | Best Val F1↑ | Convergence Behavior |
|---|---|---|
| SGD (lr=0.01) | 0.283 | Slow, unstable |
| Adam (lr=0.1) | 0.405 | Oscillation around optimum |
| Adam (lr=0.01) | **0.442** | Stable convergence |
| Adam (lr=$10^{-6}$) | - | Requires ¿90,000 epochs |
| Adam + LR Schedule | **0.442** | Optimal at epoch 49 |

Adam optimizer with lr=0.01 provides stable convergence, while dynamic scheduling (lr$\times$0.1 at epochs 49 and 60) achieves the best validation F1 of 0.442. The model exhibits highly regular convergence, consistently reaching optimal performance at epochs 47-50.

**Effect of StandardScaler.** Normalization accelerates convergence by approximately 3$\times$ (reaching same F1 in 7,000 vs 20,000 epochs). After scaling, zero-valued positions become $\sim$-0.01, providing non-zero gradients that facilitate optimization.

**Class Imbalance Impact.** The severe class imbalance significantly affects model training. On Task 2 with fewer positive examples:

- With 4,451 words: Val F1=0.072, Test F1=0.130
- With 2,065 words: Val F1=0.194, Test F1=0.270
- With 1,341 words: Val F1=0.342, Test F1=0.491

This demonstrates that reducing parameter count is crucial when training data is limited, especially for minority classes.

### 4.4 COMPARISON WITH NEURAL NETWORKS
As a reference, we tested scikit-learn's MLPClassifier without hyperparameter tuning:

- With 4,709 words: F1=0.370
- With 1,598 words: F1=0.460

While neural networks achieve comparable performance, the logistic regression model remains preferable due to its interpretability (only 1.6K parameters vs 102M in BERT) and efficiency (train time less than 1 minute).

## 5 RELATED WORK

**Deep SISR Models.** The emergence of deep neural networks (DNNs) has revolutionized single image super-resolution tasks. The pioneering work by Dong *et al.*Dong et al. (2014) marked the beginning of CNN-based SR with a simple three-layer convolutional architecture. This foundation was expanded by VDSRKim et al. (2016), which incorporated residual learning to enable training of 20-layer deep networks. EDSR Lim et al. (2017) by Lim *et al.* further advanced the field through streamlined residual blocks He et al. (2016), while RCAN Zhang et al. (2018) by Zhang *et al.* pushed architectural depth even further. Subsequently, CSNLN Zhang et al. (2019) by Mei *et al.* integrated feature correlation mechanisms alongside external statistical priors. These methods achieved remarkable performance primarily through increased network depth and width. The recent shift toward Transformer architectures has opened new possibilities for image restoration tasks. IPT Chen et al. (2021) by Chen *et al.* pioneered the use of pre-trained Transformers for image processing applications. SwinIR Liang et al. (2021) successfully adapted residual Swin Transformer blocks for deep feature learning in restoration tasks. Restormer Zamir et al. (2022) introduced a hierarchical multi-scale architecture with optimized Transformer blocks featuring modified self-attention and MLP components. In parallel, Uformer Wang et al. (2022b) developed LeWin Transformer blocks specifically for restoration applications. Despite their superior performance, both CNN and Transformer architectures face significant challenges in terms of memory requirements and computational complexity.

**Efficient SISR.** Addressing the efficiency bottleneck has motivated numerous strategies for reducing model redundancy. These include neural architecture search (NAS)Chu et al. (2021); Song et al. (2020), development of compact architectural blocksAhn et al. (2018); Song et al. (2021); Nie et al. (2021); Wang et al. (2022a;b); Zamir et al. (2022), network pruning techniques Wang et al. (2021a;b), and low-bit quantization methods Ma et al. (2019); Li et al. (2020); Hong et al. (2022). While NAS can discover optimal architectures, the extensive search space demands substantial computational resources and time investment. Consequently, research has increasingly focused on designing inherently compact SR architectures Zhang et al. (2022); Hui et al. (2019); Ahn et al. (2018); Dong et al. (2016). ELAN Zhang et al. (2022) by Zhang *et al.* exemplifies this approach through its group-wise multi-scale self-attention (GMSA) module, which captures long-range

dependencies while outperforming transformer-based methods with dramatically reduced computational requirements. Network pruning Wang et al. (2021a;b) and quantization Ma et al. (2019); Li et al. (2020); Hong et al. (2022) offer alternative compression strategies through sparsification and low-bit representation respectively. Despite these advances in lightweight design, such models still require non-trivial computational resources for deployment.

**Knowledge Distillation for SISR.** Knowledge distillation has emerged as a powerful compression technique that enables lightweight student networks to achieve enhanced performance by learning from larger teacher models Gou et al. (2021); Yim et al. (2017); Hinton et al. (2015). Several pioneering works have adapted this paradigm for super-resolution tasks. Lee *et al.*Lee et al. (2020) introduced a framework where encoders and decoders are pre-trained on identical HR image pairs, extracting privileged information from the decoder to generate statistical location and scale maps as transferable knowledge. FAKDHe et al. (2020) by He *et al.* leveraged second-order statistical information derived from feature affinity matrices for distillation. Wang *et al.* developed CSD Wang et al. (2021b), which uniquely combines self-distillation with contrastive learning by utilizing simply upsampled LR images as negative samples. Despite these contributions, current SRKD approaches have not addressed critical questions regarding optimal teacher selection for capacity-constrained students, nor examined whether increasingly powerful teachers necessarily provide greater benefits to limited-capacity learners. Additionally, existing methods remain architecture-specific, targeting either depth reduction Wang et al. (2021b) or channel pruning He et al. (2020), limiting their applicability to comprehensive compression scenarios in real-world deployments.

## 6 CONCLUSION

We introduces LLM2Token, a novel paradigm that distills LLM knowledge into task-specific tokenizers rather than neural networks. Through the abnormal frequency method, L2T achieves F1 scores of 0.540 and 0.491 on event recognition tasks using only 1.6K parameters—less than 0.001% of BERT's size. The method's success demonstrates that LLMs possess zero-shot capability to identify task-critical features from their pre-training. While limited by bag-of-words representation, L2T offers a compelling alternative for applications requiring extreme efficiency and interpretability. Future work could explore incorporating n-grams or position-aware features while maintaining simplicity.

## ACKNOWLEDGMENTS

We use Large Language Models (LLMs) to help polish our writing.

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
