# OpenReview forum: "LLM2Token: Distilling Large Language Models into Task-Specific Tokenizer"
_ICLR.cc/2026/Conference — Submitted to ICLR 2026_

### Official Review · Reviewer_DqBF · 2025-10-25

**Soundness:** 1
**Presentation:** 2
**Contribution:** 2
**Rating:** 2
**Confidence:** 4

**Summary:**

This paper presents LLM2Token (L2T), a novel knowledge distillation paradigm that departs from traditional neural network-based methods by distilling the knowledge of Large Language Models (LLMs) into task-specific tokenizers instead of smaller neural networks.

**Strengths:**

The claim that prior knowledge from LLMs can be distilled in the form of tokenizer is interesting.

**Weaknesses:**

1. First of all, this paper contains quite a few writing errors. For instance, there is a spelling mistake in the word "distilling" in Line 11; line 5 in table 3 is confusing; additionally, the paper claims in Line 15 and Line 66 that the proposed method outperforms GPT-2 and GPT-3.5 respectively, which has left me very confused.
2. This paper claims that LLM2tokenizer is a knowledge distillation method, yet it fails to compare LLM2tokenizer with mainstream knowledge distillation methods. Besides, the experiments in this paper are only conducted on a single dataset, which cannot effectively validate the claims made in the paper.
3. I am very confused by the training process shown in Figure 3—it seems extremely unstable and fluctuates constantly. The authors need to analyze this phenomenon. In my view, this reflects the uncertainty and instability of the proposed method, which makes it unable to be generalized to other tasks.
4. LLM2tokenizer relies on a large model to extract tokens, and I highly doubt whether this result is reliable across all tasks. This paper needs to conduct a detailed analysis of this issue.

**Questions:**

See Weaknesses

---

### Official Review · Reviewer_ReQ1 · 2025-10-30

**Soundness:** 2
**Presentation:** 2
**Contribution:** 2
**Rating:** 2
**Confidence:** 4

**Summary:**

The authors propose a new paradigm distillation method for LLMs, which extracts the teacher model’s knowledge into a task-specific tokenizer via feature selection, rather than distilling it into a traditional student model. A simple classifier is then trained on the resulting tokenizer to complete the distillation. This approach reduces distillation costs and makes the distilled knowledge more interpretable.

**Strengths:**

The authors propose a novel lightweight knowledge distillation paradigm, which enables extremely fast distillation.

**Weaknesses:**

1.There is a typo in the abstract, line 14: “simgle-layer” should be corrected to “single-layer”.

2.In the experimental section, the authors only compare their method with three other feature selection approaches, but do not benchmark against traditional knowledge distillation methods, which makes it difficult to convincingly demonstrate the effectiveness of the proposed method as a distillation paradigm.

3.The experiments are limited to event recognition tasks, without evaluation on a broader range of language model tasks such as generation, mathematical reasoning, or code understanding, thus limiting the generality claims of the approach.

4.The token selection relies heavily on prompts fed to the teacher model, yet the authors do not describe how these prompts are designed, nor do they analyze whether different prompt choices affect the token selection outcome.

**Questions:**

1.The authors should clarify which LLM served as the teacher during distillation. Additionally, would the choice of teacher model affect the quality of the selected tokens and the downstream performance of the proposed method?

2.Since one of the claimed advantages is interpretability, it would be helpful to visualize or list the token sets selected for different tasks to demonstrate what knowledge is being distilled.

3.A direct comparison with standard distillation baselines would help position this work more clearly and validate its effectiveness beyond feature selection.

---

### Official Review · Reviewer_h2A2 · 2025-11-01

**Soundness:** 1
**Presentation:** 2
**Contribution:** 1
**Rating:** 2
**Confidence:** 4

**Summary:**

This manuscript claims to introduce a method to distill LLMs into a "task-specific tokenizer" instead of a smaller neural net. The proposed method boils down to prompting an LLM to obtain a vocabulary suitable for a task, and then training a logistic regression model over such a vocabulary.

**Strengths:**

- The idea of distilling LLMs into non neural kinds of models is not without interest
- The goal to reduce the size and computational cost of LLMs is certainly praiseworthy

**Weaknesses:**

- The manuscript mistakes a tokenizer with a logistic regression model over a feature vocabulary. Indeed, the paper is not about tokenizers at all (i.e., algorithms that map sequences of strings in one vocabulary into sequences of strings in a different vocabulary; cf. [Sennrich et al, 2016](https://arxiv.org/abs/1508.07909), [Kudo et al, 2018](https://arxiv.org/abs/1808.06226), [Gastaldi et al, 2025](https://arxiv.org/abs/2407.11606)). What is here called "tokenizer" is actually a feature selection model for training classifiers.
- What the manuscript calls "distillation" of a "task-specific vocabulary" is just prompting an LLM, with the prompt "You are an expert in linguistics and data science. Your goal is to identify the most discriminative words for a text classification task." The "entirely new paradigm" proposed boils down to "instruct the LLM to act as an expert in linguistics and data science. We clearly define the task, including a description of the positive and negative classes." This can be barely considered a method, let alone a scientific one, under any rigorous standard.
- The remaining part of the work reduces to applying logistic regression over the obtained vocabulary, which is far from being a novel or original technique.
- In the face of these flaws, many claims concerning the novelty and advantages of the method appear as unsubstantiated. In particular, the efficiency and the interpretability are no more and no less than those of logistic regression.
- The manuscript exhibits numerous typos and linguistic and stylistic mistakes, which could, of course, be fixed, but which also suggest that the paper was not in its final state at the moment of submission.

**Questions:**

- Why do you call "tokenizer" a classifier?
- What is your justification for prompting LLMs as a rigorous scientific method?

---

### Official Review · Reviewer_Apzd · 2025-11-01

**Soundness:** 2
**Presentation:** 2
**Contribution:** 2
**Rating:** 2
**Confidence:** 4

**Summary:**

The paper proposes a “knowledge distillation” method to transfer LLM's knowledge into a “symbolic and interpretable representation”, terming it task-specific tokenizer. Instead of transferring logits or features to the student model, the LLM is prompted to generate discriminative tokens for a classification task. These tokens form a custom vocabulary used to represent inputs via one-hot encoding, followed by logistic regression model training for event recognition. Experiments on two binary event recognition tasks from the People's Daily corpus show that L2T outperforms manual, chi-square, and linear model based feature selection, achieving 0.408 F1 with 0.1% of the parameters compared to previous models. The authors argue this demonstrates LLMs' zero-shot ability to identify task-relevant features, offering an alternative to standard distillation.

**Strengths:**

1. The idea of utilizing LLM for zero-shot feature selection is conceptually interesting. It reframes LLM use beyond generation or fine-tuning, suggesting an interesting way of using pretrained knowledge to inform lightweight models.
2. The paper emphasizes efficiency, and achieves strong performance using extremely lightweight models, which could appeal to low-resource deployment settings.
3. The methodology is clearly written and can be easily followed. All steps from prompting to classification are sufficiently described.

**Weaknesses:**

1. The assumption that the LLM can provide the top-k most relevant tokens is not directly tested. It is possible that the LLM outputs semantically similar words rather than truly discriminative tokens, as the LLM is prompted with examples rather than accessing the actual data distribution. The paper would be stronger if it compared representations built from LLM-selected tokens with those derived from actual top-k tokens based on LLM's probability distribution.

2. The framing as “knowledge distillation” may be misleading, as it deviates from the standard teacher-student optimization, logit transfer, or soft-target training. The method is closer to LLM-guided feature selection. The terminology demands a stronger comparison with established distillation literature, and the lack of such discussion weakens the conceptual positioning.

3. The necessity of using an LLM for the distillation is not clear. The LLM operates only through prompting and does not access the training data directly. It remains unclear whether the same or better token sets could be obtained from TF-IDF or frequency-based selection. Including such comparisons would clarify whether the LLM meaningfully contributes beyond conventional feature extraction.

4. The experiments are limited in scope. The evaluation is confined to two binary classification tasks from a single Chinese corpus. It remains unclear whether the method generalizes to other domains, languages, or multi-class setups. The paper should include English corpus for broader relevance.

5. The paper lacks analysis of the quality and consistency of the selected tokens. There are no examples illustrating what kinds of tokens the LLM identifies, whether they are semantically meaningful, or whether results are stable across different prompts or runs. Moreover, the interpretability claim of the LLM-generated vocabulary itself is not demonstrated. Providing qualitative examples would make the interpretability argument more convincing.

**Questions:**

1. Please see the Weakness section for points where additional analysis/clarification or experimentation would strengthen the paper.

2. Could you specify which LLM(s) were used, their version or API source, and prompting parameters such as temperature and top-p?

3. Could you include the examples of few-shot inputs and LLM’s outputs?

4. Did you observe substantial synonymy or duplication among the generated tokens? If so, how was this handled?

5. The Related Work section on Deep SISR Models appears unrelated to the proposed method. Can you clarify the relevancy of it in the paper’s argument?

---

### Official Review · Reviewer_b2CE · 2025-11-01

**Soundness:** 2
**Presentation:** 2
**Contribution:** 2
**Rating:** 2
**Confidence:** 4

**Summary:**

The paper presents LLM2Token, a novel distillation paradigm that leverages large language models (LLMs) to generate a set of task-relevant keywords for text classification. By using either a task description or a small number of examples, LLM2Token efficiently extracts discriminative features that capture the underlying knowledge of the LLM. The approach is evaluated on two benchmarks and demonstrates strong performance using only a minimal number of parameters.

**Strengths:**

- Despite its simplicity and efficiency, the proposed method achieves strong performance on two binary classification benchmarks.

- The method presents a novel approach to knowledge distillation by focusing on token-level selection, offering a distinct alternative to traditional output- or feature-based distillation techniques.

**Weaknesses:**

- The distillation process is tailored to a specific task, which means a separate distillation must be performed for each new task, potentially limiting scalability and reusability.
- The evaluation is limited to two benchmarks and binary classification tasks. Given that LLMs are designed for general-purpose language understanding, comparing them with a task-specific distilled method may not provide a fully fair or comprehensive assessment of their relative capabilities.
- Since L2T is only evaluated on classification tasks, its applicability to other NLP task types—such as generation, question answering, or sequence labeling—remains unclear and warrants further investigation.

**Questions:**

Please refer to the weakness section.

---

### Meta-Review · Area_Chair_8xVk · 2026-01-02

**Summary:**

The main concern is that the proposed method is mischaracterized and over-claimed. Reviewers argue that LLM2Token is essentially LLM-guided feature selection combined with logistic regression, rather than a genuine knowledge distillation method or a tokenizer in the standard NLP sense.

**Reviewer Concerns:**

No reviewer concerns were addressed, as the authors did not submit a rebuttal.

**Reviewer Scores:**

No reviewer concerns were addressed, as the authors did not submit a rebuttal.

---

### Decision · Program_Chairs · 2026-01-26

Reject